# The Safety and Efficacy of an Unflanged 4F Pancreatic Stent in Transpancreatic Precut Sphincterotomy for Patients with Difficult Biliary Cannulation: A Prospective Cohort Study

**DOI:** 10.3390/jcm11195692

**Published:** 2022-09-26

**Authors:** Jieun Ryu, Kyu-Hyun Paik, Chang-Il Kwon, Dong Hee Koh, Tae Jun Song, Seok Jeong, Won Suk Park

**Affiliations:** 1Division of Gastroenterology, Department of Internal Medicine, Daejeon St. Mary’s Hospital, College of Medicine, The Catholic University of Korea, Daejeon 34943, Korea; 2Digestive Disease Center, CHA Bundang Medical Center, CHA University School of Medicine, Seongnam 13497, Korea; 3Research Group for Endoscopic Instruments and Stents, Korean Society of Gastrointestinal Endoscopy, Seoul 03741, Korea; 4Division of Gastroenterology, Department of Internal Medicine, Hallym University Dongtan Sacred Heart Hospital, Hallym University College of Medicine, Hwaseong 18450, Korea; 5Division of Gastroenterology, Department of Internal Medicine, Ulsan University College of Medicine, Asan Medical Center, Seoul 05505, Korea; 6Division of Gastroenterology, Department of Internal Medicine, Inha University School of Medicine, Incheon 22332, Korea

**Keywords:** ERCP, pancreatitis, endoscopic sphincterotomy, stent

## Abstract

Prophylactic pancreatic stenting effectively reduces the rate and severity of post-ERCP pancreatitis (PEP) in the precut technique; however, studies on the optimal type and duration of the stent are still lacking. This prospective study evaluated the incidence and severity of PEP and the rate of spontaneous stent dislodgement in patients undergoing transpancreatic precut sphincterotomy (TPS) accompanied by prophylactic pancreatic stenting with an unflanged plastic stent (4F × 5 cm) for difficult biliary cannulation. A total of 247 patients with naïve papilla were enrolled in this study, and data were collected prospectively. In the final analysis, 170 and 61 patients were included in the standard cannulation technique and TPS groups, respectively. The incidence of PEP in the standard cannulation technique and TPS groups was 3.5% and 1.6% (*p* = 0.679), respectively. The technical success rate of selective biliary cannulation in the TPS group was 91.8%. The spontaneous dislodgement rate of the prophylactic plastic stent was 98.4%. In conclusion, an unflanged pancreatic stent (4F × 5 cm) placement in TPS for patients with failed standard cannulation technique is a safe and effective measure due to low adverse events and few additional endoscopic procedures for removing the pancreatic duct (PD) stent.

## 1. Introduction

Endoscopic retrograde cholangiopancreatography (ERCP) is the preferable procedure for treating biliary tract diseases; however, it is frequently associated with serious post-ERCP complications. Post-ERCP pancreatitis (PEP) is the most common of these complications and can occasionally be fatal in severe cases. Selective biliary cannulation is key in performing endoscopic retrograde cholangiography (ERC) without complications, particularly PEP. However, several large studies have identified failure of selective biliary cannulation as an independent risk factor for PEP [1,2,3,4], occurring in 8.2–49.5% [5,6,7,8,9,10]. Therefore, various cannulation techniques have been developed and evaluated to prevent PEP by overcoming difficult biliary cannulation [11].

If selective biliary cannulation using the standard cannulation technique (SCT) fails, many endoscopists prefer an early switch to the needle-knife precut (NKP) technique because of concerns that prolonged attempts of SCT will increase the incidence of PEP and delay the procedure. The NKP techniques are representative of NKP sphincterotomy and fistulotomy and showed a lower rate of PEP when compared with prolonged attempts of SCT in meta-analyses [12,13]. In one randomized controlled trial (RCT), the complication rate of NKP fistulotomy was statistically lower than that of NKP sphincterotomy and was reported to be safer [14].

The pancreatic duct guidewire (PGW) placement is a technique for gaining access to the bile duct in patients in whom SCT fails, with the double guidewire technique (DGT) and TPS representatively used [5]. Since the first description of DGT by Dumonceau et al. [15] in 1998, several studies [5,6,7,16,17] have shown its efficacy and safety in treating difficult biliary cannulation. TPS, first reported by Goff [18] in 1995, was initially developed for application in various pancreatic diseases [19,20,21] but is now widely used as a rescue technique for patients with difficult biliary cannulation. In a meta-analysis, TPS had a higher cannulation success rate than NKP papillotomy, with no difference in the rate of PEP, which has been demonstrated as an effective and safe procedure for managing difficult biliary cannulation [22].

Whether increased PEP rate is related to the precut technique, repeated, or prolonged papillary manipulation, or repeated pancreatic cannulation remains unclear. However, the sole use of the above-mentioned advanced techniques results in a higher incidence of PEP than SCT in unselected patients [23,24,25,26,27]. Several studies have demonstrated that prophylactic insertion of a pancreatic duct (PD) stent effectively lowers the incidence and severity of PEP [28,29,30,31,32,33,34]. Therefore, it is recommended that most advanced cannulation techniques be accompanied by prophylactic PD stenting to reduce the incidence of PEP [29,35,36]. 

Nevertheless, studies on the optimal types and insertion duration of a PD stent for PEP prevention are lacking. In addition, an additional endoscopic procedure for removing the PD stent may be required, as it causes considerable burden and inconvenience to the operator and patient. There is no study on how to induce safe spontaneous dislodgement without complications. Here, we hypothesized that TPS would facilitate opening to the bile duct in patients with difficult biliary cannulation, reduce the incidence of PEP when accompanied by prophylactic PD stenting, and induce spontaneous dislodgement of the prophylactic PD stent. Therefore, this prospective study aimed to evaluate the incidence of PEP and the rate of spontaneous stent dislodgement in patients undergoing TPS accompanied by prophylactic PD stenting with an unflanged PD stent (4F × 5 cm).

## 2. Materials and Methods

### 2.1. Patients and Study Design

This single-center, prospective cohort study was approved by the institutional review board of the Daejeon St. Mary’s Hospital, the Catholic University of Korea. Between September 2018 and June 2021, a total of 353 ERCP procedures were performed on patients with naïve major papilla. After excluding patients with pre-ERCP hyperamylasemia, acute or chronic pancreatitis, pancreatic cancer, tumors of the major duodenal papilla, coagulation disorders, and history of bile duct surgery, 247 patients were enrolled, and their data were collected prospectively. Standard wire-guided biliary cannulation was performed in 180 patients, and selective biliary cannulation through a TPS was attempted in 67 patients with difficult biliary cannulation. Finally, we analyzed 61 patients in the TPS group and 170 patients in the SCT group, excluding cases of unintentional instrument insertion or contrast injection into the PD, when the procedures did not meet the study protocol, or when patients were lost to follow-up (Figure 1).

### 2.2. Definition and Outcome Measurement 

The SCT was defined as guidewire-assisted biliary cannulation without contrast injection using a sphincterotome or standard ERCP cannula [36]. TPS was defined as a technique that facilitates access to the bile duct through a septal incision between the pancreatic and bile ducts using a standard sphincterotome on a guidewire deeply placed in the PD [36]. We defined difficult biliary cannulation as the following: (1) >10 min spent on selective biliary cannulation, and (2) more than one unintentional cannulation into PD [36,37].

Based on a consensus document proposed by Cotton et al. [38], mild PEP was defined as a new or worsened abdominal pain, serum amylase level of ≥3 times the upper normal limit for more than 24 h, and requirement for added admission for 2–3 d. Moderate PEP was defined as clinical pancreatitis requiring 4–10 d hospitalization. Severe PEP was defined as cases requiring more than 10 d hospitalization, presence of local complications, or need for drainage or surgery [39]. The presentation of post-sphincterotomy bleeding (PSB) was classified as clinically significant or insignificant based on the clinical evidence of gastrointestinal bleeding and a decrease in hemoglobin level [40]. Occult gastrointestinal bleeding with a decrease in hemoglobin level of <3 g/dL was defined as clinically insignificant PSB. Clinically significant PSB was graded as mild, moderate, or severe according to a consensus document proposed by Cotton et al. [38]. Mild bleeding was defined as clinically overt bleeding and a decrease in hemoglobin level of <3 g/dL. Moderate bleeding was defined as requiring <4 units of transfusion and no interventions. Severe bleeding was defined as requiring >5 units of transfusion or interventions [38,39].

The primary outcome parameter was the rate of PEP in patients undergoing a TPS. The secondary outcome parameters were the rate of other ERCP-related complications such as post-ERCP hyperamylasemia, PSB, perforation, the overall success rate of selective biliary cannulation, and the rate of spontaneous dislodgement of PD stent in patients undergoing a TPS. Laboratory tests were performed the day before the procedure, 4 h after the procedure, and in the morning of the day after the procedure. Abdominal x-rays were followed on days 1 and 2, 2 weeks, and 2 months after the ERCP procedure to evaluate the dislodgement of the prophylactic PD stent.

### 2.3. Endoscopic Procedures

All ERCP procedures were performed under balanced conscious sedation with a titrated dose of midazolam, pethidine, and propofol. Opioid analgesics, anticholinergics, protease inhibitors, and antibiotics were administered appropriately before and after the procedure. Two high-volume endoscopists who perform more than 200 ERCPs yearly [41] performed all procedures using an Olympus video duodenoscope (JF-260V; Olympus, Tokyo, Japan). A standard “traction-type” sphincterotome (CleverCut3™; Olympus, Tokyo, Japan) preloaded with a guidewire (0.025-inch VisiGlide 2; Olympus, Tokyo, Japan) and ENDO CUT I current mode in ERBE electrosurgical generator (VIO^®^ 300 D; ERBE Elektromedizin, Tübingen, Germany) was used for cannulation and sphincterotomy. Prophylactic PD stent placement used a plastic stent (Advanix^TM^ Pancreatic Stent; Boston Scientific, Marlborough, MA, USA) with a single pigtail on the duodenal side, a diameter of 4F, a length of 5 cm, and without an internal flange.

In all patients, the SCT was always performed as the initial procedure. In difficult biliary cannulation, TPS was attempted as a primary rescue procedure if successful PGW placement was achieved. In failed selective biliary cannulation with TPS, NKP techniques were performed as alternative rescue procedures. The TPS procedure is as follows (Figure 2): First, the tip of a standard sphincterotome mounted on a guidewire was inserted into the PD and advanced a little into the orifice of the PD. Second, sphincterotomy was performed in the bile duct direction at an 11 o’clock position and exposure of the bile duct lumen [7,42]. Third, leaving the first guidewire deeply in the PD, the guidewire-assisted biliary cannulation was performed using a sphincterotome with an additional guidewire. All patients who underwent TPS were accompanied by prophylactic PD stenting.

### 2.4. Statistical Analysis

Statistical analysis was performed using Statistical Package for the Social Sciences (SPSS, Version 18.0; Chicago, IL, USA). Continuous variables are represented as mean ± standard deviation (SD), and categorical variables are represented as numbers and proportions (%). Comparison between groups was carried out using the independent sample t-test for continuous variables and χ2-test with or without Fisher’s exact test for categorical variables. *p*-values < 0.05 were considered statistically significant. 

We hypothesized that the incidence of PEP in TPS with prophylactic PD stent placement would not be inferior to the previously reported results of SCT. We estimated the sample size based on the study of Sofuni A. et al., which indicated that the incidence rate of PEP was 3.2% in patients with prophylactic PD stent placement. The expected number of patients was 64, with an alpha value of 0.05, a power of 90%, and a drop-out rate of 10%.

## 3. Results

### 3.1. Patients’ Characteristics

Of the 247 patients enrolled in the study, 180 (72.9%) achieved selective biliary cannulation with the SCT and 67 (27.1%) did not. Of the 67 patients with difficult biliary cannulation, TPS was performed in 64 patients (95.5%) who had successful PGW placement within 10 min. NKP papillotomy was performed in three (4.5%) patients who had failed PGW placement. Finally, 170 ERCP procedures were classified as the SCT group, and 61 ERCP procedures were classified as the TPS group (Figure 1). The patients in the two groups were comparable in their baseline characteristics, with no statistically significant difference in any of these parameters (Table 1).

### 3.2. Post-ERCP Complications

As shown in Table 2, the incidence of post-ERCP hyperamylasemia was significantly high in the TPS group (52.5% vs. 28.8%, *p* = 0.001). However, the overall incidence of PEP was statistically insignificant (1.6% vs. 3.5%, *p* = 0.679). Six of seven patients with PEP experienced mild grades, and all patients were discharged without an extension of their planned hospital stay. In the SCT group, one patient with PEP experienced a moderate grade with a 4d extension of hospital stay. There were no severe grades of PEP in either group.

The overall incidence of PSB was statistically insignificant between the TPS and SCT groups (6.6% vs. 7.6%, *p* = 1.00). Only three patients in the SCT group developed clinically significant PSB, and none in the TPS group. Of the three patients, two had mild bleeding that did not require a blood transfusion, and one had moderate bleeding requiring transfusion of two packed RBCs. Post-ERCP perforation and related death did not occur in the two groups. 

### 3.3. Technical Outcomes

The application rates of SCT, TPS, and NKP are shown in Figure 1. The success rate of SCT, the primary cannulation technique for selective biliary cannulation, was 72.9% (180/247 patients). The remaining 67 patients (27.1%) were classified as the group with difficult biliary cannulation. The PGW placement for TPS was successful in 64/67 patients (95.5%). In these 64 patients, the success rate of selective biliary cannulation was 92.2% (59 patients). Three patients who had failed PGW placement and five who had failed selective biliary cannulation after TPS underwent NKP techniques as an alternative rescue procedure. Selective biliary cannulation was achieved in seven out of eight patients (87.5%).

The mean cannulation times for SCT, TPS, and NKP were 3.2 ± 2.0 min, 9.4 ± 11.5 min, and 31.9 ± 12.0 min, respectively. The mean ERCP times were 19.5 ± 1.4 min, 26.3 ± 12.9 min, and 41.8 ± 11.8 min, respectively. There was a significant difference in time between each group according to the study protocol. In patients who underwent successful TPS as a primary rescue procedure, the time from failed SCT to selective biliary cannulation was 6.0 ± 10.6 min, and the mean total cannulation time was 9.4 ± 11.5 min. In patients who underwent successful NKP as an alternative rescue procedure for failed TPS, the time from failed TPS to selective biliary cannulation was 10.9 ± 9.6 min, and the mean total cannulation time was 31.9 ± 12.0 min (Table 3).

### 3.4. Dislodgement Rate 

A total of 64 patients underwent PD stenting after TPS. Of these sixty-four patients, two had undergone endoscopic removal of prophylactic PD stent at the 2nd ERCP session of 2 d intervals, and one was lost to follow-up, which resulted in the exclusion of three patients from the analysis. The overall spontaneous dislodgement rate of 4F PD stents in the TPS group was 98.4% (60/61 patients). The highest rates of spontaneous dislodgement occurred between 2 and 14 d. The dislodgement rate was 65.6% (40/61 patients). The rate of spontaneous dislodgement by period was 8.2% (5/61) within 2 d, 65.6% (40/61) within 2 weeks, and 24.6% (15/61) after 2 weeks. One patient (4.0%) experienced failed spontaneous dislodgement, which was removed by endoscopy after 2 months (Table 4). There were no complications related to early stent dislodgement within 24 h or delayed stent dislodgement over 2 weeks.

## 4. Discussion

Since the advent of ERCP, selective biliary cannulation has remained a prerequisite for successful ERC and is one of the most technically challenging points. To date, selective biliary cannulation fails in 15–35% of cases with SCT alone, even when performed by experienced endoscopists [33,43]. In this study, the success rate of the primary SCT was 72.9%, and the incidence of difficult biliary cannulation was 27.1%, which was similar to the results of previous studies [33,43].

Various advanced cannulation techniques have been developed to overcome difficult biliary cannulation. Of these, the NKP is the most commonly used technique for accessing the bile duct, and has a high success rate; however, it also has high risks of ERCP-related complications, which are reported in 2–34% of patients [23,34,44,45,46,47,48]. In addition, the NKP techniques have a steep learning curve for inexperienced endoscopists and are often technically challenging, even for experienced endoscopists. In contrast, the advantage of TPS is that the sphincterotome does not need to be changed to a needle-knife device, and the depth and direction of the incision are better controlled using a traction-type sphincterotome compared to a needle-knife device [49,50]. In a meta-analysis of six RCTs, TPS increased the success rate of selective biliary cannulation compared with persistent SCT, and early NKP techniques are superior to other techniques in decreasing PEP rates [17]. Furthermore, several studies have demonstrated that TPS resulted in a higher rate of selective biliary cannulation compared with the PGW-assisted technique, early NKP techniques, and the PD stent-assisted technique [17,22,51]. Based on these studies, TPS should be considered the most suitable option in terms of increasing the success rate and avoiding PEP in cases of difficult biliary cannulation. Therefore, we attempted TPS as a primary rescue procedure for patients with difficult biliary cannulation and considered the NKP technique as an alternative rescue procedure if TPS failed. In this study, the cannulation of primary SCT achieved a 72.9% (180/247) success rate, and in failed cases, the sequential use of TPS increased the selective biliary cannulation rate to 96.8% (239/247). When TPS failed, applying an alternative rescue procedure using NKP techniques increased the overall success rate to 99.6% (246/247). The time from failed SCT to selective biliary cannulation in TPS was 6.0 ± 10.6 min, and the mean total cannulation time was 9.4 ± 11.5 min. The mean total SCT cannulation time of 3.2 ± 2.0 min was acceptable to endoscopists. Given these results, the use of TPS as a rescue procedure should be considered an effective option to increase the success rate of selective biliary cannulation in patients with difficult biliary cannulation.

The overall incidence of PEP is approximately 5–15% and 0.1–0.8% in severe cases [2,3,50,52,53]. Difficult biliary cannulation is widely accepted as an independent PEP risk factor [52]. An increased number of cannulation attempts, extended duration of cannulation, unintentional PGW placement, pancreatic injection of contrast media, and thermal injury by precut sphincterotomy are factors that can increase the risk of PEP in difficult biliary cannulation [1,24,50,54]. A meta-analysis has shown that the incidence of PEP in patients who underwent precut sphincterotomy was 5.28%, compared with 3.10% (*p* < 0.001) in patients who did not [24,36]. Several studies have reported that the incidence of PEP was 5.5–21% [55,56,57] in TPS alone and 3.5–4.1% [50,58,59,60] in prophylactic PD stent placement after TPS. Moreover, the most recent studies have demonstrated that prophylactic PD stenting reduced the incidence of PEP [28,60,61,62,63,64]. In addition, the ESGE clinical guideline strongly recommends prophylactic PD stenting in patients with PGW-assisted techniques, such as TPS [29,35,36,65]. In this study, the incidence of post-ERCP hyperamylasemia in the TPS group was significantly higher than that in the SCT group (52.5% vs. 28.8%, respectively; *p* < 0.001). The incidence of PEP in the TPS group was slightly lower than that in the SCT group (1.6% vs. 3.5%, respectively; *p* = 0.679). However, the incidence of PEP between the two groups was statistically insignificant. Therefore, we can infer that prophylactic PD stenting prevented the aggravation of PEP, although TPS increased the rate of post-ERCP hyperamylasemia caused by difficult biliary cannulation, placement of PD guidewire, and thermal injury by sphincterotomy. These results suggest that TPS, accompanied with prophylactic PD stenting, in difficult biliary cannulation is as safe as SCT in unselected patients in terms of PEP. 

For prophylactic PD stents, previous studies have demonstrated that stents of 5F diameter are more effective than 3F stents in preventing PEP [66,67,68]. Although one RCT reported a lower PEP rate with 5F stents of 3 cm length than those of 5 cm [69], another study has been controversial [70]. Endoscopists generally prefer to use stents without an internal flange to facilitate spontaneous dislodgement [71], and with a flange or pigtail on the duodenal side to prevent intraductal migration [29,72]. However, some experts recommend using a flanged pancreatic stent to avoid early stent dislodgement after TPS [36] because the stent needs to be left in place for at least 12–24 h to prevent PEP [73]. In one RCT, PD stents kept for more than 2 weeks were associated with delayed PEP [67]; however, in another study, this association was not found [74]. To date, the placement duration, diameter, length, and use of PD stents with or without internal flange to prevent PEP in TPS have not been fully evaluated. The induction of spontaneous dislodgement of the PD stent at an appropriate time after TPS is a considerable issue, as additional endoscopic procedures for its removal can lead to a burden and risk for both endoscopists and patients. In this study, we used a plastic stent of diameter 4F and 5 cm in length, with a single pigtail on the duodenal side, and without an Internal flange to induce spontaneous dislodgement in patients with TPS. The overall spontaneous dislodgment rate of the PD stent was 98.4%, and 73.8% were removed within 2 weeks, which was similar to a previous study [71]. In addition, there was no early stent dislodgement within 12 h. These results suggest that TPS will not increase PEP due to early dislodgement of the pancreatic stents.

In conclusion, our study demonstrated that TPS accompanied with prophylactic PD stenting for difficult biliary cannulation was safe in terms of reducing PEP, and was effective in increasing the success rate of selective biliary cannulation in patients with difficult biliary cannulation with few needs for endoscopic removal of the prophylactic PD stent. Therefore, endoscopists should first consider the TPS with prophylactic PD stenting as an effective rescue technique for difficult biliary cannulation.

## Figures and Tables

**Figure 1 jcm-11-05692-f001:**
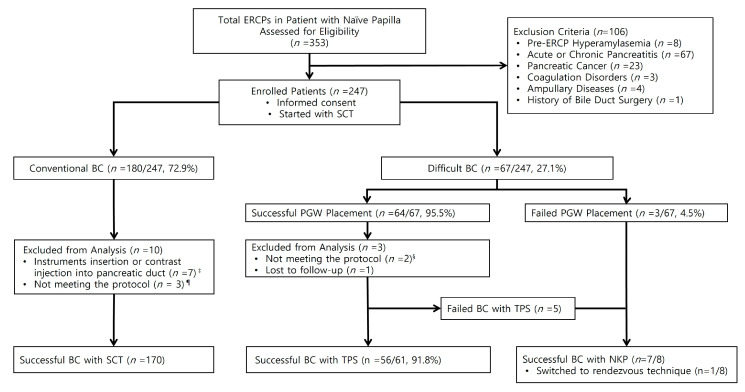
The selection of participants in the study. **Legend**. **ERCP**: endoscopic retrograde cholangiopancreatography, **SCT**: standard cannulation technique, **BC:** Biliary Cannulation, **PGW:** Pancreatic Duct Guidewire, **TPS**: transpancreatic precut sphincterotomy, **NKP**: needle-knife precut. ^‡^ Intentional or unintentional insertion of the instruments or contrast injection into the pancreatic duct. ^¶,§^ Protocol violence. The endoscopic removal of prophylactic pancreatic stent was performed during the 2nd ERCP session which occurred within 2 days.

**Figure 2 jcm-11-05692-f002:**
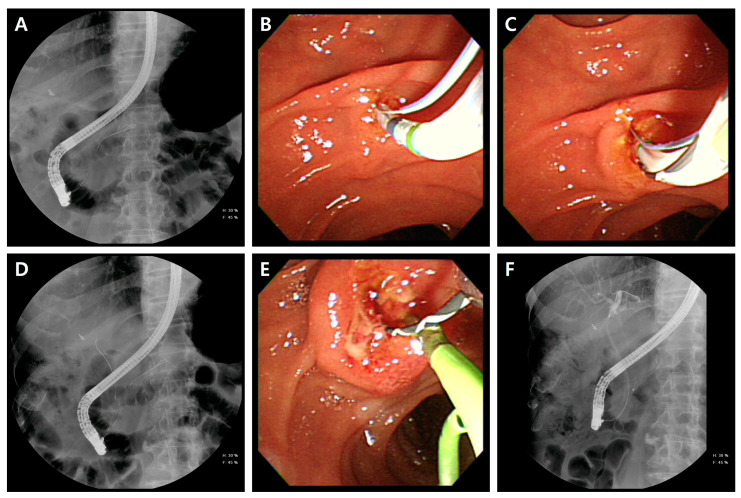
Transpancreatic sphincterotomy technique. (**A**) Fluoroscopy image showing the guidewire inserted in the pancreatic duct. (**B**) The septum was cut with a sphincterotome from the pancreatic duct towards the bile duct axis. (**C**) The cannulation toward the bile duct was performed while leaving the guidewire in the pancreatic duct. (**D**) Selective bile duct cannulation was successfully achieved. (**E**) The placement of pancreatic duct stent. (**F**) After prophylactic pancreatic stenting, termination of all procedures.

**Table 1 jcm-11-05692-t001:** Clinical characteristics of patients undergoing ERCP.

Data	SCT, *n* (%)	TPS, *n* (%)	*p*-Value
Gender			0.939
Male Female	91 (53.5) 79 (46.5)	33 (54.1) 28 (45.9)	
Age, years (mean ± SD)	69.21 ± 15.81	68.98 ± 15.88	0.989
BMI, kg/m^2^ (mean ± SD)	24.21 ± 3.92	24.45 ± 4.49	0.556
Indications			0.337
Cholelithiasis BD Cancer/stricture	138 (81.2) 32 (18.8)	46 (75.4) 15 (24.6)	
Comorbidities			
Diabetes Mellitus Hypertension Chronic Liver Disease Chronic Kidney Disease History of Cholecystectomy History of Pancreatitis Antiplatelet Medication	48 (28.2) 86 (50.6) 11 (6.5) 8 (4.7) 15 (8.8) 1 (0.6) 51 (30.0)	15 (24.6) 29 (47.5) 3 (4.9) 1 (1.6) 2 (3.3) 1 (1.6) 13 (21.3)	0.583 0.683 1.00 0.451 0.251 0.459 0.193
Laboratory Findings			
Hemoglobin, g/dL (mean ± SD) hsCRP, mg/dL (mean ± SD) Total bilirubin, mg/dL (mean ± SD) ALT, IU/L (mean ± SD) Creatinine, mg/dL (mean ± SD) Amylase, U/L (mean ± SD) Lipase, U/L (mean ± SD)	12.66 ± 1.77 4.49 ± 5.95 4.17 ± 5.19 197.14 ± 212.70 1.06 ± 1.35 61.34 ± 25.62 42.42 ± 28.99	12.98 ± 1.49 3.26± 5.37 3.99± 4.64 168.82 ± 180.91 0.87 ± 0.24 65.59 ± 27.83 43.00 ± 27.83	0.201 0.158 0.809 0.355 0.266 0.181 0.893
Periampullary Diverticulum	64 (37.6)	19 (31.1)	0.364
Image findings			
Diameter of PD, mm (mean ± SD) Diameter of BD, mm (mean ± SD)	2.16 ± 1.65 13.49 ± 5.31	2.01 ± 1.08 14.31 ± 5.17	0.524 0.304

**Legend**. **ERCP**: endoscopic retrograde cholangiopancreatography, **SCT**: standard cannulation technique, **TPS**: transpancreatic precut sphincterotomy, **SD**: standard deviation, **BMI**: body mass index, **BD**: bile duct, **PD**: pancreatic duct.

**Table 2 jcm-11-05692-t002:** Complication rates of SCT and TPS.

Complications	SCT, *n* (%)	TPS, *n* (%)	*p*-Value
Post-ERCP Hyperamylasemia	49/170 (28.8%)	32/61 (52.5)	0.001 *
Overall PEP ^†^	6/170 (3.5)	1/61 (1.6)	0.679
Mild	5/170 (2.9)	1/50 (1.6)	
Moderate	1/170 (0.6)	0/50 (0.0)	
Severe	0/170 (0.0)	0/50 (0.0)	
Overall PSB	13/170 (7.6)	4/61 (6.6)	1.00
Clinically Insignificant	10/170 (5.9)	4/61 (6.6)	0.765
Hemostatic Procedure	3/170 (1.8)	0/61 (0.0)	0.568
Clinically Significant ^‡^	3/170 (1.8)	0/61 (0.0)	0.568
Mild Moderate Severe	2/170 (1.2) 1/170 (0.6) 0/170 (0.0)	0/61 (0.0) 0/61 (0.0) 0/61 (0.0)	
Perforation	0/168 (0.0)	0/61 (0.0)	NA

**Legend**. **ERCP**: endoscopic retrograde cholangiopancreatography, **SCT**: standard cannulation technique, **TPS**: transpancreatic precut sphincterotomy, **PEP**: post-ERCP pancreatitis, **PSB**: post-sphincterotomy bleeding. * *p*-values < 0.05 was accepted as statistically significant. ^†^ Clinically significant PEP was graded as mild, moderate, or severe post-ERCP pancreatitis according to a consensus document proposed by Cotton et al. ^‡^ Clinically significant PSB was graded as mild, moderate, or severe post-sphincterotomy bleeding according to a consensus document proposed by Cotton et al.

**Table 3 jcm-11-05692-t003:** Technical outcomes of the patients with difficult biliary cannulation.

	SCT	Difficult Cannulation	
		TPS	NKP
Success Rate for Deep SBC, *n* (%)	180/247 (72.9)	59/64 (92.2)	7/8 (87.5)
Total ERCP Procedure Time, min (mean ± SD)	19.5 ± 11.4	26.3 ± 12.9	41.8 ± 11.8
Total Cannulation Time, min (mean ± SD)	3.2 ± 2.0	9.4 ± 11.5	31.9 ± 12.0
Time to Selective BC ^†^, min (mean ± SD)	3.2 ± 2.0	6.0 ±10.6	10.9 ± 9.6

**Legend**. **ERCP**: endoscopic retrograde cholangiopancreatography, **SCT**: standard cannulation technique, **TPS**: transpancreatic precut sphincterotomy, **NKP**: needle-knife precut, **SBC**: selective biliary cannulation, **BC**: Biliary Cannulation. ^†^ Time from successful or failed TPS to selective biliary cannulation.

**Table 4 jcm-11-05692-t004:** The rate of spontaneous dislodgement of prophylactic pancreatic stents.

	*n*	(%)
Successful Spontaneous Dislodgement	60/61	98.4
Within 12 h	0/61	0.0
Within 48 h	5/61	8.2
Within 2 weeks	40/61	65.6
Within 8 weeks	15/61	24.6
Failed Spontaneous Dislodgement	1/61	4.0

## Data Availability

Data are available on request because of restrictions, e.g., privacy or ethics.

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
