# Peer review of "The Safety and Efficacy of an Unflanged 4F Pancreatic Stent in Transpancreatic Precut Sphincterotomy for Patients with Difficult Biliary Cannulation: A Prospective Cohort Study"

_jcm, 2022, doi:10.3390/jcm11195692_

Round 1

Reviewer 1 Report

This study examines the efficacy and adverse effects of transpancreatic sphincterotomy in subjects for whom standard cannulantions techniques fail.

1- They say prophylactic pancreatic stenting effectively prevents PEP. However it is not so: "Prophylactic pancreatic stenting reduces the rate and severity of PEP" reads better.

2- Prophylactic pancreatic stenting is not a rescue technique, but a prevantive measure.

3- They say "Selective biliary cannulation is key in performing ERCP. This is ERC not ERP.

4- Pancreatic duct guide-wire placement is not a novel techinique anymore.

5- Not "Treating biliary cannulation", "Managing biliary cannulation".

6- In text they say they included 180 SCT and 67 TPS patients and they finally analyzed 170 TPS and 61 SCT patients ???? Incongruity !

7- What is the definition of obscure gastrointestinal bleeding ? It is not clear.

8- With which method did they ascertain the dislodgement time of pancreatic stents ?

9- "Different cannulation techniques" better reads than "Advanced ....."

10- There is no mention of efficacy of rectal indomethacin in the prevention of PEP in Discussion.

Author Response

Point 1: They say prophylactic pancreatic stenting effectively prevents PEP. However, it is not so: "Prophylactic pancreatic stenting reduces the rate and severity of PEP" reads better.

Response 1: Thank you. Following your comment, 1] Line 20-21 has been revised: Prophylactic pancreatic stenting effectively reduces the rate and severity of post-ERCP pancreatitis (PEP) in the precut technique. 2] Lines 22-25 have been revised: This prospective study evaluated the incidence and severity of PEP and the rate of spontaneous stent dislodgement in patients undergoing TPS accompanied by prophylactic pancreatic stenting with unflanged plastic stent (4F x 5 cm) for difficult biliary cannulation. 3] Line 335 has been revised: I replaced the word “preventing” with “reducing”.

Point 2: Prophylactic pancreatic stenting is not a rescue technique, but a prevantive measure

Response 2: Thank you for pointing this out. Following your comment, Lines 30-33 have been revised: In conclusion, an unflanged pancreatic stent (4F x 5 cm) placement in TPS for patients with failed standard cannulation technique is a safe and effective measure due to low adverse events and few additional endoscopic procedures for removing the pancreatic duct (PD) stent.

Point 3: They say "Selective biliary cannulation is key in performing ERCP. This is ERC not ERP.

Response 3: Thank you for pointing this out. Following your comment, 1] Line 40-42 has been revised: Selective biliary cannulation is key in performing endoscopic retrograde cholangiography (ERC) without complications, particularly PEP. 2] Line 259-260 has been revised: Since the advent of ERCP, selective biliary cannulation has remained a prerequisite for successful ERC and is one of the most technically challenging points.

Point 4: Pancreatic duct guide-wire placement is not a novel techinique anymore.

Response 4: Thank you for this. Following your comment, Line 54-56 has been revised: The pancreatic duct guidewire (PGW) placement is a technique for gaining access to the bile duct in patients in whom SCT fails, with the double guidewire technique (DGT) and TPS representatively used.

Point 5: Not "Treating biliary cannulation", "Managing biliary cannulation".

Response 5: Thank you for your comment. Following your comment, Line 60-63 have been revised: In a meta-analysis, TPS had a higher cannulation success rate than NKP papillotomy with no difference in the rate of PEP, which has been demonstrated as an effective and safe procedure for managing difficult biliary cannulation.

Point 6: In text they say they included 180 SCT and 67 TPS patients and they finally analyzed 170 TPS and 61 SCT patients ???? Incongruity !

Response 6: We are sorry for the confusion. Following your comments, Lines 92-93 have been revised: Finally, we analyzed 61 patients in the TPS group and 170 patients in the SCT group,

Point 7: What is the definition of obscure gastrointestinal bleeding? It is not clear.

Response 7: We are sorry for the confusion. Following your comment, Lines 114-116 have been revised: I replaced the word “Obscure” with “Occult” for clarity. Occult gastrointestinal bleeding with a decrease in hemoglobin level of <3 g/dL was defined as clinically insignificant PSB.

Point 8: With which method did they ascertain the dislodgement time of pancreatic stents ?

Response 8: This is described in lines 126-128: Abdominal x-rays were followed on days 1 and 2, 2 weeks, and 2 months after the ERCP procedure to evaluate the dislodgement of the prophylactic PD stent.

Point 9: "Different cannulation techniques" better reads than "Advanced ....."

Response 9: Advanced cannulation technique refers to secondary procedures performed when the standard cannulation technique fails and is used in many papers. In contrast to the standard cannulation technique, there is a sentence where the word "advanced" rather than "different" is appropriate to distinguish it. Following your comment, on lines 44 (various advanced cannulation techniques) and 277 (other advanced techniques), the word "advanced" has been deleted.

Point 10: There is no mention of efficacy of rectal indomethacin in the prevention of PEP in Discussion.

Response 10: Rectal indomethacin was not used in the patients included in our study. Also, since the main topic of this paper is whether it is possible to reduce the incidence and severity of PEP by using TPS and unflanged plastic stents in rescue from difficult cannulation, we could not describe it.

Reviewer 2 Report

Thank you for the opportunity to review the manuscript "The Safety and Efficacy of an Unflanged 4F Pancreatic Stent in Transpancreatric Precut Sphincterotomy for Patients with Difficult Biliary Cannulation: A Prospective Cohort Study". 

This prospective cohort study analyzes the incidence of post-ERCP pancreatitis/PEP, ERCP-associated complications, the success rate of selective biliary cannulation, and the rate of spontaneous dislocation of a pancreatic stent in patients undergoing transpancreatic precut sphincterotomy/TPS (= difficult biliary cannulation) and use of an unflanged pancreatic duct (PD) stent (4F x 5 cm). The treatment results are compared with those of patients with standard cannulation technique/SCT (conventional biliary cannulation/BC). 

The authors demonstrated a very low incidence of PEP compared to the SCT group (overall PEP: TPS = 1.6 %; SCT = 3.5 %; p = 0.679). This proved the safety and prophylactic effect to avoid PEP when using the investigated PD stent. For the PD stent used, the spontaneous dislodgment rate was 98.4%, with no spontaneous dislodgments within 12 hours postintervention.

The study investigates a medically highly relevant topic. Primary and secondary endpoints are clearly defined. The design of the study as well as the methodology used are comprehensible. All results are well presented. In the discussion section, the results are discussed at a very good level and taking into account the current literature on the subject.

Only note for correction: page 2, lines 92/93: here the number of patients for the groups SCT and TPS is interchanged.

Author Response

Point 1: Only note for correction: page 2, lines 92/93: here the number of patients for the groups SCT and TPS is interchanged.

Response 1: Thank you for your observation. Following your comments, Lines 91-92 have been revised: Finally, we analyzed 61 patients in the TPS group and 170 patients in the SCT group,

Round 2

Reviewer 1 Report

The suggested revisions were done. Publishable